# Contextualizing goal preferences in fear-avoidance models. Looking at fatigue as a disabling symptom in fibromyalgia patients

**Cecilia Peñacoba**[1]*, **Irene López-Gómez**[1], **Maria Angeles Pastor-Mira**[2], **Sofía López-Roig**[2], **Carmen Ecija**[1]

**1** Department of Psychology, Rey Juan Carlos University, Alcorcón, Madrid, Spain, **2** Department of Behavioral Sciences and Health, Miguel Hernández University, Campus de Sant Joan, Alicante, Spain

* cecilia.penacoba@urjc.es

## Abstract

The fear-avoidance model provides an explanation for the development of chronic pain, including the role of perception (i.e. pain catastrophism) as an explanatory variable. Recent research has shown that the relationship between pain catastrophism and avoidance is influenced in turn by different psychological and contextual variables, highlighting the affective-motivational ones. From this perspective, the Goal Pursuit Questionnaire (GPQ) was developed to measure the preference for hedonic goals (mood-management or pain-avoidance goals) over achievement goals in musculoskeletal pain patients. Recently, the Spanish version of the GPQ in fibromyalgia patients has been validated. Our aim has been to adapt the Spanish version of GPQ from pain to fatigue symptoms and to validate this new questionnaire (GPQ-F) in fibromyalgia. Despite the recognition of fibromyalgia as a complex disorder and the need for a differential study of its symptoms, fatigue, despite its high prevalence and limiting nature, remains the forgotten symptom. We conducted a cross-sectional study with 231 women with fibromyalgia. Previously, we adapted the Spanish GPQ for fatigue symptoms with three sub-studies (group structured interview, self-administration questionnaire and thinking-aloud; n = 15–27 patients). We explored the GPQ structure and performed path analyses to test conditional mediation relationships. Exploratory factor analysis showed two factors: 'Fatigue-avoidance goal' and 'Mood-management goal' (39.3% and 13.9% of explained variance, respectively). The activity avoidance pattern fully mediated the relation between both catastrophizing and fatigue-avoidance goals with fatigue. The study shows initial findings about the usefulness of the GPQ-F as a tool to analyze goal preferences related to fatigue in fibromyalgia. The results supported the mediational role of activity avoidance patterns in the relationship between preference for fatigue-avoidance goals and fatigue.

## Introduction

The understanding of fibromyalgia (FM) has evolved from the early concept of a solely pain based condition, to the acceptance of symptoms beyond pain, including fatigue, emotional

**Funding:** This study was funded by the Health Research Fund (Fondo de Investigación en Salud) from the Instituto de Salud Carlos III (Spain) co-financed by the European Union through the Fondo Europeo de Desarrollo Regional (FEDER), Grant Number PI17/00858. Maria Angeles Pastor-Mira and Sofía López-Roig contribution was supported by a research grant from MINECO (PSI2016-79566-C2-1-R). The funders had no role in study design, data collection and analysis, decision to publish, or preparation of the manuscript.

**Competing interests:** The authors have declared that no competing interests exist.

disorders, sleep disturbance, and other somatic symptoms, as exemplified by the most recent criteria [1, 2]. The burden of illness for FM is considerable, with personal suffering, wide-reaching psychosocial implications of compromised function at home and work, and considerable direct and indirect costs [3]. FM is associated with reduced physical function, constituting one of the most problematic outcomes for fibromyalgia patients [4, 5]. In fact, improving the physical function of these patients has become one of the prime objectives [6].

The fear-avoidance model (FA) provides an explanation for the development of chronic pain [7]. FA emphasizes the importance of the beliefs patients hold about their pain and their role in promoting disabling fear and avoidance. Specifically, in the FA model, pain initiates a set of cognitive, emotional and behavioural responses that may or may not exacerbate pain and disability [8]. At the core of the FA model is how patients interpret pain, which is why pain catastrophizing has been widely studied within these models. A catastrophic misinterpretation of pain leads to an excessive fear of pain/injury which in turn leads to the avoidance of physical activity what contributes to exacerbate pain and disability [9]. The previous associations find evidence in the fact that pain catastrophizing is one of the most widely investigated and robust psychological predictors of poor outcomes in fibromyalgia, in particular, and chronic pain in general [10, 11].

However, recent research has shown that the association between pain catastrophism and avoidance is not linear, but rather a complex relationship influenced in turn by different psychological and contextual variables [12]. In this context, two types of variables have aroused the interest of researchers due to their possible explanatory role of the effects of catastrophizing on physical impact in models of fear of movement. Specifically, affective-motivational models [13] postulate that moods and signal goal attainment and may influence these relationships. The significance of fear and avoidance, within a broad motivational and emotional context, has been largely ignored [8, 14].

Individuals who catastrophize pain may primarily adopt hedonic goals aimed at avoiding the threat of pain, rather than performing the task [15]. It has been hypothesized that both long-term achievement goals and short-term hedonic goals could be associated to increased pain and disability, predominantly in patients with high negative affect [16]. In fact, there is evidence of high levels of negative affect in FM patients [17, 18], which positively associate with activity avoidance [19], catastrophizing impact [20], fatigue [21], and functional limitation [20, 22].

In this context, taking into account this affective-motivational perspective, Karsdorp and Vlaeyen (2011) [23] developed the Goal Pursuit Questionnaire (GPQ) to measure the extent to which participants preferred hedonic goals (mood-management or pain-avoidance goals), over achievement goals in various situations in a sample of musculoskeletal pain patients. The GPQ contains two reliable subscales; one measuring a person's preference for mood-management goals in relation to achievement goals (mood-management goal subscale) and another measuring a person's preference for pain-avoidance goals in relation to achievement goals (pain-avoidance goal subscale). Participants who strongly endorsed pain-avoidance goals also reported higher pain and disability levels while controlling for biographical variables and pain catastrophizing. Goal pursuit and negative affect were found to be independently related to disability. Recently, Pastor-Mira et al. [20] developed the Spanish version of the GPQ finding the same subscales than Karsdorp and Vlaeyen [23], although in their model, as a novel aspect, the mediating role of activity patterns in the relationship between goals and health outcomes was included. Preference for pain avoidance goals was found to always be related to pain, disability and fibromyalgia impact through activity patterns [20].

Despite some findings providing support for the validity of an affective-motivational approach to chronic pain, little research has examined how a goal-based motivational

construct may influence fatigue symptoms in individuals with chronic illnesses. As we have pointed out, in spite of pain often being the main symptom in the diagnosis of fibromyalgia, fatigue has been shown to be highly prevalent and persistent in these patients [24–29]. Although more that 75% of FM patients report fatigue [30], conceptualizing it as one of their most concerning symptoms that impact on quality of life [31, 32], little is known regarding the psychosocial variables involved in its maintenance.

Focusing on fatigue and based on the studies of Karsdorp and Vlaeyen and Pastor-Mira et al. [20, 23], the aim of the present study has been to adapt the Spanish version of the Goal Pursuit Questionnaire (GPQ) from pain to fatigue symptoms and to validate this new questionnaire, the GPQ-F, in a sample of individuals with FM. A second aim has been to test whether hedonic and achievement goals, measured with the GPQ-F are related to fatigue, pain and disability in individuals with FM and whether these relationships are mediated by activity patterns, while controlling for negative affect and pain catastrophizing. It was hypothesized that a strong endorsement of hedonic goals (fatigue-avoidance and mood-management goals) would be related to greater fatigue, pain and FM impact and that this relationship would be mediated by the activity patterns. To the best of our knowledge, this is the first study focusing on fatigue to test these hypotheses.

## Materials and methods

### Design and procedure

A descriptive, cross-sectional study was carried out. The study was approved by the Bioethics Committee of Rey Juan Carlos University (Reference 160520165916; PI17/00858) and all participants signed an informed consent form to take part in the project.

### Participants

Two hundred and thirty one women with FM diagnosis according to the American College of Rheumatology (ACR) criteria (Wolfe et al., 1990, 2010) participated in this study. Inclusion criteria for this study were: have a fibromyalgia diagnosis, female, age over 18 years. Patients were recruited from different fibromyalgia associations in Spain (Madrid, Ciudad Real, Albacete, Guadalajara, and Toledo) in which FM diagnosis is a mandatory requirement for association. A minimum n was established at 200, following the established criteria for factor analyses [33, 34].

Participants age ranged between 30 and 78 years old, with a mean age of 56.91 (SD = 8.94). The majority of the sample was married or living with a partner (78.8%). Regarding education level, 13.8% had not finished elementary school, 52.6% had finished elementary school, 26.6% had finished high school and 6.9% had finished college or university studies. From the total of women, 32.9% were working at home, 32% were retired, 12.4% were working outside home, 12.6% were unemployed and 10% were on sick leave at the time the study was developed. The mean time elapsed since they were diagnosed was 12.14 years (SD = 8.45) (range 1–46 years) and the mean time suffering from fatigue problems was 19.73 years (SD = 13.95).

### Variables and instruments

Socio-demographic and clinical variables were measured with an "ad hoc" questionnaire. The battery of questionnaires included the following instruments:

Goal Pursuit Questionnaire (fatigue) (GPQ-F): An adaptation of the Spanish version [20] of the Goal Pursuit Questionnaire (GPQ) [23] was created to measure the extent to which participants preferred hedonic goals (mood-management or fatigue-avoidance goals), over

achievement goals in various situations. For the adaptation of the GPQ-F, the Spanish GPQ [20] was taken as a starting point. The Spanish GPQ [20] measures the goal pursuit of people with pain, taking into account achievement or hedonic goals which can be activated at the same time in one situation. Similarly to the original GPQ [23], the Spanish GPQ showed a structure of two factors, named: 'Pain-avoidance goal' (Factor I, 8 items, alpha = 0.90) and 'Mood-management goal' (Factor II, 6 items; alpha = 0.81). Higher mean scores in each factor indicate stronger preferences for a hedonic goal in comparison to an achievement goal, that is, to avoid pain (Factor I) or to maintain positive mood (Factor II).

For measuring goal pursuit in relation to fatigue as a symptom, we adapted the Spanish GPQ [20]. Like the Spanish GPQ, the GPQ-F contains 16 items in a 6 point Likert scale (1 = strongly disagree, 6 = strongly agree), higher scores indicate stronger preferences for a hedonic goal (fatigue avoidance or mood management) in relation to an achievement goal. The items of the Spanish GPQ were adapted to fatigue symptoms and show different situations related to work, study or leisure conflicting with achievement and hedonic goals. No situation was modified regarding the Spanish GPQ. For this adaptation, the same procedure used by Pastor-Mira et al. [20] was followed; in particular, in a field study with three sub-studies, we performed: (1) a group structured interview after group self-administration of the GPQ-F (n = 24); (2) a thinking-aloud study (n = 15); (3) a group, self-administration questionnaire comprising only the activities listed in the GPQ to study their frequency in the daily life of fibromyalgia patients (n = 27), and in the case of the items related to fatigue avoidance, the perception that these activities were actually associated with fatigue. If not, patients were asked to describe another activity with similar fatigue consequences. These sub-studies aimed to assess the feasibility of the GPQ-F and its clarity (instructions, items and answer scale). Following the original instructions, participants had to imagine "as vividly as possible" the situation presented in one vignette and rate their agreement with a specific thought that showed preference for achievement or hedonic goals in that specific situation.

Pain catastrophizing: The total score of the Spanish adaptation of the Pain Catastrophizing Scale (PCS) [35] was used to measure pain catastrophizing. This scale contains 13 items answered in a 5-point Likert scale from 0 (not at all) to 4 (all the time). Scores range from 0 to 52 and higher scores represent higher catastrophizing ($\alpha$ = 0.94).

Fatigue: The "General fatigue" subscale of the Multidimensional Fatigue Inventory (MFI) [36] was used. It is a 20-item self-report instrument designed to measure fatigue including the following dimensions: general fatigue, physical fatigue, mental fatigue, reduced motivation and reduced activity. Items are scored on a Likert scale ranging from 1 to 5. Cronbach's alpha for the present study was 0.72.

Pain intensity: Measured with the mean score of the maximum, minimum, and usual pain intensity during the last week and pain intensity at time of the assessment [37]. These items were answered with an 11-point numerical rating scale (0 = "no pain at all" and 10 = "the worst pain you can imagine") ($\alpha$ = .85). High mean scores indicate high pain intensity. Cronbach's alpha for the present study was 0.72.

Fibromyalgia impact: The Spanish version of the Fibromyalgia Impact Questionnaire-Revised (FIQ-R) [38] was used. It consists of 21 items with an 11-point Likert response (from 0 to 10) format that evaluates three associated domains: physical function, overall impact and symptoms. Fibromyalgia impact was measured with the total score of the questionnaire (rank 0–100). Items are answered on an 11 point numerical rating scale from 0 to 10, with different verbal anchors depending on the item. Higher scores represent higher disability or higher impact perception. Cronbach's alpha for the present study was 0.92.

Activity patterns: For conceptual reasons (affective-motivational models of fear of movement), two patterns were the selected: activity avoidance and task-contingent persistence. For

the assessment of these patterns, the corresponding subdimensions of the Spanish version of the Activity Patterns Scale (APS) [19] were administered. The "Activity avoidance" subdimension contains 3 items (avoidance refers to the patients' condition of being in pain rather than the fluctuating pain experience) and the "Task contingent persistence" subdimension contains 3 items (patients persist in finishing tasks or activities despite pain). All items are scored using a 5-point Likert scale (0 = Never, 4 = Always). Cronbach's alpha for the present study was 0.72 for activity avoidance and 0.77 for task-contingent persistence.

Negative affect: The negative affect dimension of the Spanish version of the Positive and Negative Affect Schedule (PANAS) [39] was used. The PANAS questionnaire contained 10 items to assess positive affect (reflects the level to which a person feels active, enthusiastic and alert) and 10 items to assess negative affect (is a state of general distress and unpleasurable engagement). Items are rated on a 5-point Likert scale from 1 (not at all or very slightly) to 5 (extremely). Cronbach's Alpha for the present study was 0.87.

## Data analysis

First, our data were analyzed to clarify whether they fitted the conditions for linear factor analysis [40]. We tested the floor and ceiling effects of each item (percentage of response above 95% in scores 1 and 6). Secondly, for the validity analysis based on internal structure, we conducted an Exploratory Factor Analysis (EFA) using the maximum likelihood (ML) method and oblique rotation following the recommended standards [40]. After selected factors by the scree plot, Kaiser' rule and baseline theory, we obtained the Kaiser-Meyer-Olkin index and the Bartlett sphericity test to explore the sampling and data adequacy. We retained items with loading values greater than 0.45. Item-corrected scale correlation and correlations of factors with other constructs were performed with the Pearson coefficient for assessing the validity of the GPQ-F. Statistical significance was set at $p < 0.05$. To test the internal consistency of the scales in our sample we calculated Cronbach's alpha and Omega index. The data were analyzed with the SPSS-24.

Structural equation models were performed by the lavaan package in R [41] and figures were generated by the lavaanPlot package in R [42]. The MVN package in R [43] was used to study assumptions of multivariate and univariate normality. Mardia's multivariate normality test showed no multivariate normality. The Shapiro-Wilk univariate normality tests showed non-normality in all the variables, except for the Negative affect variable. No missing data were found. Outliers were detected by the outliers R package [44], established on the adjusted quantile method based on Mahalanobis distance, and substituted by the median value.

Structural equation models tested include one mediator (activity avoidance pattern or task-contingent persistence pattern), depending on the model, and four independent variables (fatigue-avoidance goals, mood-management goals, negative affect and pain catastrophizing). The dependent variables were fatigue, pain and fibromyalgia impact. The parameter estimation was calculated by maximum likelihood estimation with robust standard errors and a Satorra-Bentler scaled test statistic, due to the non-normality of the data. A fit-criteria assessment was conducted according to the Hu and Bentler study [45] using the Root Mean Square Error of Approximation (RMSEA), the Standardized Root Mean Residual (SRMR), the Comparative Fit Index (CFI), and the Tucker Lewis Index (TLI) fit indices. A ratio of $\chi 2/df < 2$ suggests an acceptable fit. An RMSEA size below 0.06 suggests a well-fitting model. A CFI and TLI above 0.95 indicate a good fit. An SRMR of less than 0.09 also indicates a good fit.

Fig 1 represents the tested structural models, with exogenous and endogenous variables. All were observed variables and measured on an interval rating scale. The arrows in the figures indicate the hypothesized relationships among variables. Fig 1 represents two large models to be tested: one of them with activity avoidance as mediator and the second with task contingent

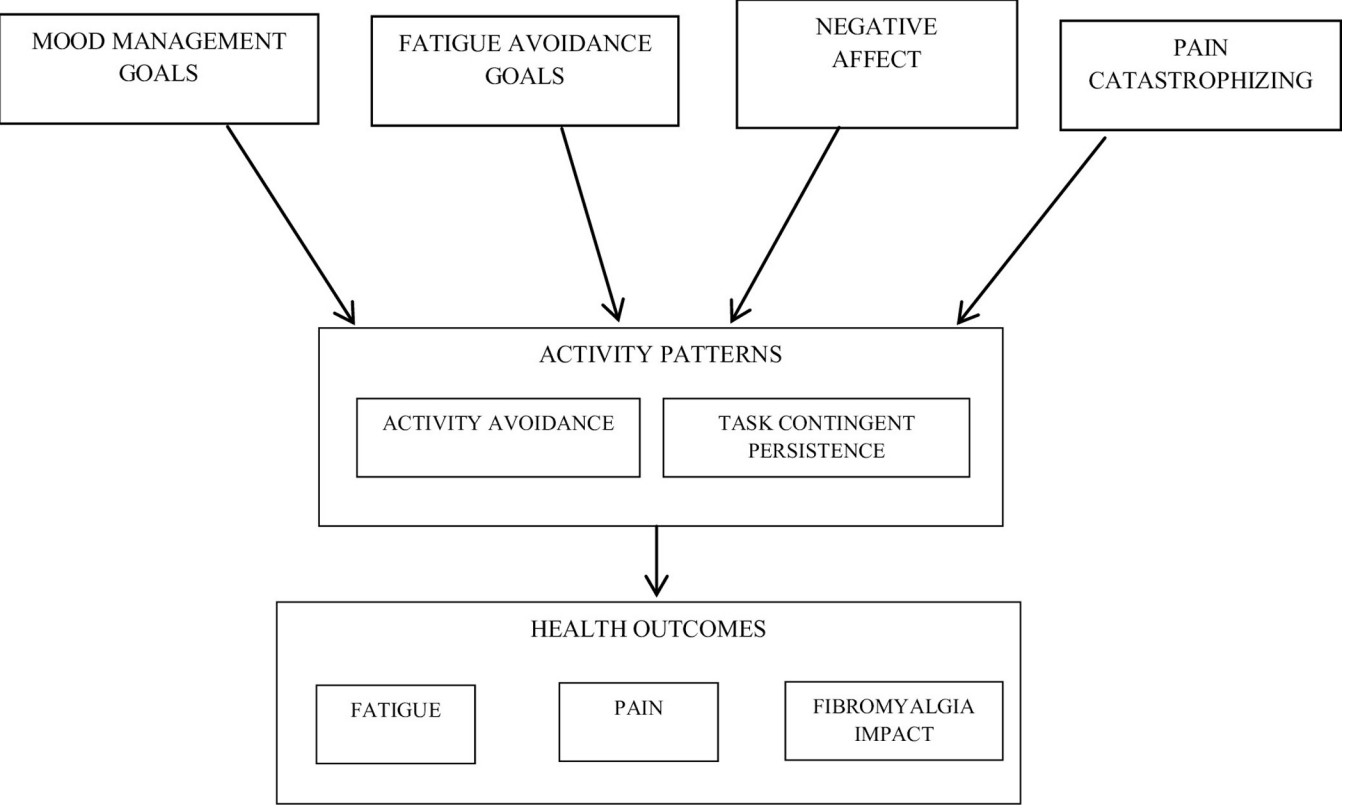

**Fig 1. Tested structural model with activity patterns (activity avoidance and task contingent persistence).**

persistence as mediator. These models were tested with four independent variables (fatigue-avoidance goals, mood-management goals, negative affect and pain catastrophizing) and with three different dependent variables (fatigue, pain and fibromyalgia impact). Therefore, 6 analyses were performed.

## Results

### GPQ-F analysis

In all the items, all the answer options (six) obtained some percentage. Likewise, a normal distribution was found for all the items (Kolmogorov-Smirnoff test). No floor or ceiling effects were found. The highest skewness (0.9) was found in item number 4. We found a KMO and Bartlett index of 0.89 and 1363.4 (p<0.001) respectively, which guarantees to perform an EFA, regarding the adequacy of the sample and the correlation matrix. Two major factors and one minor factor are shown in scree plot analysis. Given that they did not reach the minimum stablished factor loading and/or they were high loadings on different factors, items 1, 5, 8, and 16 were removed from the scale. Thus, a new EFA was carried out excluding the above items (1, 5, 8, and 16) (KMO = 0.87; Bartlett test = 952.6; $p < 0.001$), showing two major factors and explaining a 43.9% of the variance. Factor I ('Fatigue-avoidance goal': 39.3% of explained variance; six items) relates to the choice between fatigue-avoidance goals or achievement goals in different situations, with high scores reflecting stronger preferences for fatigue-avoidance goals. Factor II ('Mood-management goal': 13.9% of explained variance; six items) refers to the choice between mood-management goals or achievement goals; higher scores reflect stronger

preferences for mood-management goals. A moderate correlation was found between both factors ($r = 0.47$, p<0.01). Cronbach's alpha for the Fatigue-avoidance goal factor was 0.86 and 0.76 for the mood-management goal. Table 1 shows the factor pattern matrix with loadings and descriptive data of the items.

Table 2 displays the descriptive data and correlations between variables. Results showed that the Fatigue-avoidance goal factor was positively related to the activity avoidance pattern ($p \leq 0.01$), and negatively related to the task-contingent persistence pattern ($p \leq 0.01$). The Mood-management goal factor was related to less task-contingent persistence ($p \leq 0.01$) and negatively associated to negative affect (p $\leq$ 0.05). The Fatigue-avoidance goal factor was not significantly associated to any of the outcomes from the study. The Mood-management goal factor was positively related to pain (p $\leq$ 0.01).

## Model fit

The structural equation models were designed according to Fig 1. These models include GPQ-F subscales, negative affect and pain catastrophism as predictors of activity patterns (activity avoidance and task-contingent persistence patterns) and health outcomes. All models tested showed a good fit (*see* Table 3) and are detailed below.

## Models with mediation of activity avoidance pattern

Fatigue was directly predicted by negative affect. Additionally, the activity avoidance pattern fully mediated the relationship between fatigue-avoidance goals and pain catastrophizing with fatigue (Fig 2A).

Pain intensity was directly predicted by mood-management goals and pain catastrophizing. The activity avoidance pattern partially mediated the relationship of pain catastrophizing with pain. Furthermore, this pattern fully mediated the relationship of fatigue-avoidance goals and pain (Fig 2B).

Fibromyalgia impact was directly predicted by negative affect and pain catastrophizing. The activity avoidance pattern partially mediated the relationship between pain catastrophizing

**Table 1. Item and explorative factor analysis, descriptive and internal consistency of the GPQ-F (Goal Pursuit Questionnaire for Fatigue).**

| Item | I think it is more important. . . | Loading | $M$ [a] | $SD$ | Sk | K | $r_{I-T}$ | α/Omega |
|---|---|---|---|---|---|---|---|---|
| | **Factor I. Fatigue-avoidance goal** | | **25.8** | **7.0** | -.6 | -.07 | | **.86/.89** |
| 7 | . . . for my fatigue to be reduced now, than for the house to be cleaned | .80 | 4.5 | 1.4 | -.9 | -.1 | .73 | .82 |
| 3 | . . . for my fatigue to be reduced now, than the for windows to be cleaned | .73 | 4.2 | 1.6 | -.5 | -.8 | .67 | .83 |
| 12 | . . .. for my fatigue to be reduced now, than for the sewing to be finished | .71 | 4.2 | 1.5 | -.6 | -.5 | .65 | .83 |
| 6 | . . .for my fatigue to be reduced now, than for the shopping to be finished | .70 | 4.0 | 1.6 | -.4 | -.9 | .64 | .84 |
| 11 | . . . for my fatigue to be reduced now, than for the album to be completed | .67 | 4.3 | 1.4 | -.7 | -.2 | .61 | .84 |
| 14 | . . . for my fatigue to be reduced now, than for the car to be cleaned | .66 | 4.4 | 1.5 | -.8 | -.2 | .61 | .84 |
| | **Factor II. Mood-management goal** | | **18.3** | **6.2** | .17 | -.05 | | **.75/.89** |
| 9 | . . .to decrease my boredom, than to organize clothes for laundry | .66 | 2.8 | 1.6 | .5 | -.8 | .56 | .70 |
| 10 | . . .to write a nice message (e-mail or WhatsApp) reply, than to finish the task | .61 | 2.9 | 1.5 | .3 | -.9 | .50 | .72 |
| 4 | . . .to read the exciting book now, than to finish the report on time | .58 | 2.4 | 1.5 | .8 | -.3 | .49 | .72 |
| 15 | . . .to enjoy the TV programme, than to finish my duties | .57 | 3.6 | 1.4 | -.1 | -.9 | .49 | .72 |
| 2 | . . .to tell my holiday stories or something amazing, than to finish my work | .56 | 2.7 | 1.5 | .4 | -.8 | .48 | .72 |
| 13 | . . . to have interesting conversations now, than to have the decisions made | .52 | 3.5 | 1.5 | -.0 | -.1 | .45 | .73 |

*Note*: Sk, Skewness; K, Kurtosis;

[a] Rank [1–6].

**Table 2. Pearson correlation coefficients and descriptive statistics for measured variables in the study.**

| Measure | 1 | 2 | 3 | 4 | 5 | 6 | 7 | 8 | 9 |
|---|---|---|---|---|---|---|---|---|---|
| 1. Fatigue avoidance goal | 1 | | | | | | | | |
| 2. Mood management goal | 0.47 ** | 1 | | | | | | | |
| 3. Pain catastrophism | 0.07 | 0.05 | 1 | | | | | | |
| 4. Negative affect | -0.10 | -0.13* | 0.54** | 1 | | | | | |
| 5. Activity avoidance | 0.17** | 0.10 | 0.49** | 0.34* | 1 | | | | |
| 6. Task-contingent persistence | -0.51** | -0.23** | -0.12 | 0.03 | -0.24** | 1 | | | |
| 7. Fatigue | 0.07 | -0.08 | 0.31** | 0.32** | 0.41** | -0.09 | 1 | | |
| 8. Fibromyalgia impact | 0.07 | 0.11 | 0.54** | 0.41** | 0.51** | -0.07 | 0.47** | 1 | |
| 9. Pain | 0.07 | 0.23** | 0.36** | 0.16* | 0.33** | 0.02 | 0.20** | 0.56** | 1 |
| Mean | 25.88 | 18.30 | 31.80 | 29.81 | 7.45 | 6.74 | 16.91 | 72.35 | 7.15 |
| SD | 7.09 | 6.21 | 11.68 | 8.53 | 2.71 | 2.59 | 2.90 | 17.00 | 1.52 |
| Skewness | -0.62 | 0.27 | -0.21 | -0.07 | -0.21 | -0.05 | -1.10 | -0.96 | -0.59 |
| Kurtosis | -0.03 | -0.11 | -0.75 | -0.62 | -0.26 | -0.09 | 1.36 | 0.95 | 0.86 |

**p ≤ .01.
*p ≤ .05.

with fibromyalgia impact. Moreover, this pattern fully mediated the relationship between fatigue avoidance goals with fibromyalgia impact (Fig 2C).

In sum, the activity avoidance pattern fully mediated the relation between both catastrophizing and fatigue-avoidance goals with fatigue. This pattern fully mediated the association between fatigue-avoidance goals with pain and fibromyalgia impact whereas it only partially mediated the relationship between catastrophizing with pain and fibromyalgia impact.

## Models with mediation of task-contingent persistence pattern

Fatigue was directly predicted by fatigue-avoidance goals and negative affect. the task-contingent persistence pattern was directly and negatively predicted by fatigue-avoidance goals but had no mediational role in this model (Fig 2D).

**Table 3. Fitted models with test statistics.**

| Model | $\chi^2$ (df) | CFI | TLI | RMSEA | SRMS |
|---|---|---|---|---|---|
| **Models with mediation of activity avoidance pattern** | | | | | |
| Fatigue | 3.439 (2) | 0.994 | 0.953 | 0.056 | 0.031 |
| | $p = 0.179$ | | | | |
| Pain | 2.728 (2) | 0.997 | 0.976 | 0.040 | 0.028 |
| | $p = 0.256$ | | | | |
| Fibromyalgia impact | 3.394 (2) | 0.995 | 0.965 | 0.055 | 0.031 |
| | $p = 0.183$ | | | | |
| **Models with mediation of task-contingent persistence pattern** | | | | | |
| Fatigue | 3.439 (2) | 0.993 | 0.949 | 0.056 | 0.032 |
| | $p = 0.179$ | | | | |
| Pain | 2.728 (2) | 0.997 | 0.975 | 0.040 | 0.028 |
| | $p = 0.256$ | | | | |
| Fibromyalgia impact | 3.394 (2) | 0.995 | 0.961 | 0.055 | 0.032 |
| | $p = 0.183$ | | | | |

*Note*: CFI, Comparative Fit Index; TLI, Tucker Lewis Index; RMSEA, Root Mean Square Error of Approximation; SRMR, Standardized Root Mean Square Residual.

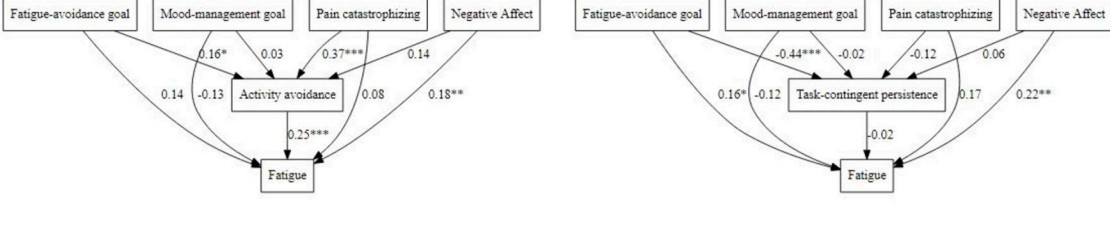

a.Model with mediation of activity avoidance patterns on fatigue

d.Model with mediation of task-contingent persistence patterns on fatigue

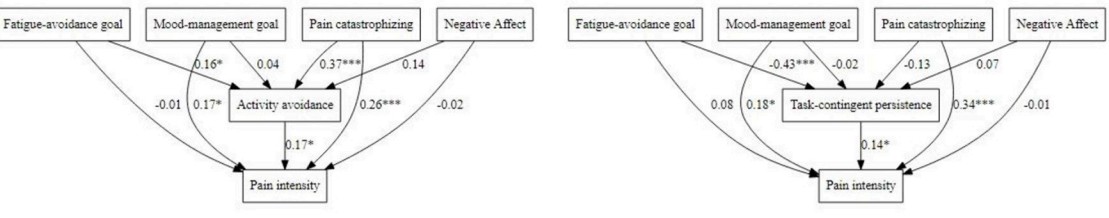

b.Model with mediation of activity avoidance patterns on pain intensity

e.Model with mediation of task-contingent persistence patterns on pain intensity

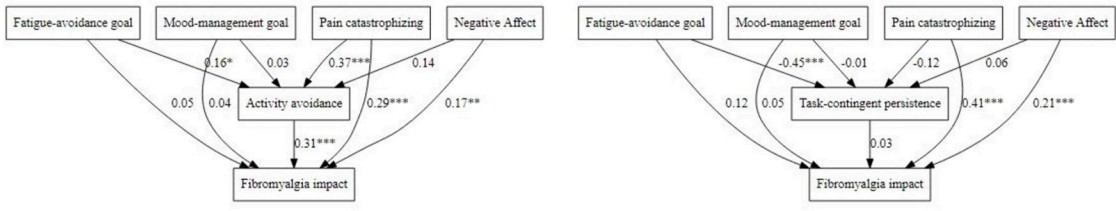

c.Model with mediation of activity avoidance patterns on fibromyalgia impact

f.Model with mediation of task-contingent persistence patterns on fibromyalgia impact

**Fig 2. Models with mediation of activity patterns (activity avoidance and task-contingent persistence) on fibromyalgia outcomes (fatigue, pain intensity and fibromyalgia impact).**

Pain intensity was directly predicted by mood-management goals and pain catastrophizing. The task-contingent persistence pattern fully mediated the relation between fatigue-avoidance goals and pain (Fig 2E).

In the case of fibromyalgia impact, it was directly predicted by negative affect and pain catastrophizing. The task-contingent persistence pattern was directly and negatively predicted by fatigue-avoidance goals but again no mediational role was found (Fig 2F).

## Discussion

The goal of this study has been to develop and validate the adaptation of the Spanish version of the GPQ [20] in relation to fatigue, an essential and forgotten symptom of FM [24]. The

analyses concerning the internal structure of the GPQ-F showed that this version is consistent with the original scale and reaches comparable psychometric standards. The EFA revealed that a two factor solution was optimal with Factor I: Fatigue-avoidance goal, and Factor II: Mood-management goal. This is the same structure showed by Pastor-Mira et al. [20] in the Spanish adaptation of the GPQ for pain, from which this GPQ-F was adapted.

Consistent with previous studies on the GPQ [20, 23], both factors showed adequate internal consistency, with the smallest values being for the mood-management goal factor. The Pearson's product-moment correlation between both subscales was moderate and equivalent to the ones found in previous studies [20, 23]. These results suggest that, in the three GPQ versions, the subscales measured different, although interrelated, constructs.

The analyses conducted also revealed good construct validity of the GPQ-F in a sample of FM patients. Significant associations were found with activity patterns, health outcomes and affect. The relationships between the GPQ-F subscales and the activity patterns were equivalent to the Spanish version of GPQ for pain [20]. Fatigue-avoidance goals were positively associated with activity avoidance, but mood-management goals did not show any significant associations. This result, along with the strong and positive relationship found between pain catastrophizing and activity avoidance, supports the fear avoidance models. These models consider that catastrophic misinterpretations of pain and preference for pain avoidance goals enhance the pain-related fear that predicts avoidance of painful activities that, in turn, increase pain and disability [20, 46]. In the present study, preference for fatigue-avoidance goals seems to play the same role than preference for pain-avoidance goals in the original Spanish version. Preference for fatigue-avoidance goals evokes fear that predicts avoidance of fatiguing activities. Activity avoidance also appears to be associated to increases in fatigue, pain and fibromyalgia impact, as shown by the medium and strong correlations found.

In the case of persistence patterns, negative and significant relationships were found between the GPQ-F subscales and task-contingent persistence, similar to what was found in the Spanish validation of the GPQ [20]. It can be concluded that a higher tendency to have hedonic goals (and, in particular, to avoid fatigue) is associated to less persistence, as goal pursuit theories state [47]. The GPQ-F and affect were found to be related only in the case of the Mood-management goal subscale. In this study, no significant association between pain catastrophizing and goal preferences was found. Pastor-Mira et al. [20] found the same result with the GPQ for pain and, as these authors mentioned, it can be explained because pain catastrophizing is measured with no motivational context whereas the GPQ items describe a context to evoke goal competition. Additionally, the pain catastrophizing measure used includes the dimensions of magnification, rumination and helplessness that may have different relationships with the GPQ subscales and, as a consequence, may limit the total correlation score [48]. Consistently with previous GPQ studies [20, 23], the GPQ subscales did not show significant associations with the outcomes studied, except for the small correlation found between pain and the Mood-management goal subscale. These results are in line with our hypothesis, as we stated that the relationship between goals and outcomes could be mediated by activity patterns. As expected, strong and medium positive correlations were found between activity avoidance patterns and the health outcomes studied, but not with task-contingent persistence [20]. It is also worth noting that fatigue was positively related to pain and fibromyalgia impact, supporting previous literature [49].

The structural equation models performed showed a good fit. It was hypothesized that a strong endorsement of hedonic goals would be related to greater fatigue, pain and fibromyalgia impact and that this relationship would be mediated by activity patterns. However, this has been only confirmed between preference for fatigue-avoidance goals and fatigue, a relationship that was fully mediated by the activity avoidance patterns. This result extends the fear avoidance

models [46, 50, 51] to fatigue. The differential results found for fatigue show the importance of behavioral patterns to explain the link between motivation and symptoms and, at the same time, highlight the need to explore fatigue as an outcome variable in its own right [24]. Although that was the only mediation found, preference for fatigue-avoidance goals was associated with activity avoidance patterns in all models tested. Simultaneously, activity avoidance always associated with fatigue, pain and fibromyalgia impact. These results confirm previous findings about the association of avoidance patterns with negative health outcomes [19, 52].

Pain catastrophizing showed a direct path in increasing activity avoidance, pain and fibromyalgia impact, supporting previous results [53]. Additionally, pain catastrophizing showed an indirect path in increasing pain, fibromyalgia impact and also fatigue through activity avoidance patterns, providing further evidence for the fear avoidance models. In the models tested, negative affect directly predicted fatigue and fibromyalgia impact, but not pain intensity. These results are consistent with previous findings [40, 54, 55]. These evidences contradict the MAI model [16, 23, 56] in our study sample and may be in line with the "discounting hypothesis", which states that when there is an obvious source to which mood changes can be attribute (as in chronic pain syndromes); mood does not moderate the relationship between goal preferences and activity patterns [57].

Karsdorp and Vlaeyen [23] recommended exploring the role of persistence patterns as a mediator between goals and health outcomes following goal pursuit theories, that state that achievement goals elicit more task persistence than hedonic goals [47]. This recommendation, along with the results showing that task-contingent persistence predicts better functioning and fewer symptoms than other types of persistence [19, 52, 53], guided our hypothesis. We hypothesized that task-contingent persistence would mediate the relationship between goal preferences and health outcomes, although our results disconfirmed it. Task-contingent persistence patterns were not a mediator of any of the relationships studied. Nevertheless, a negative relationship between fatigue-avoidance goals and task-contingent persistence patterns was found in all performed models. These results support the good functioning of the Fatigue-avoidance goal subscale of the GPQ-F, as it proves that when a subject with FM prefers to avoid fatigue they're going to persist less in the task. Another result in line with the fear avoidance models is the fact that pain catastrophizing was not significantly related with persistence patterns, contrary to the significance of the relationship found with avoidance patterns. Unexpectedly, negative affect was not significantly related to task-contingent persistence, in contrast with previous research that has found, however, contradictory results concerning the sign of this relationship [19, 20, 52]. Therefore, further studies are needed to clarify the relationship between negative affect and task-contingent persistence patterns. Although the maladaptive role of avoidance patterns has been clearly corroborated in previous literature [19, 52, 53, 58], it could be hypothesized that the role of the task persistence pattern should be interpreted from models of psychological flexibility. Based on these models, the intrinsically adaptive or maladaptive nature of certain activity patterns has been questioned [59], advocating the influence of context on the relationship between activity patterns and results [60]. In this context, recent research has pointed to the contextual role of persistence patterns [61]. In this same direction, our results, incorporating the motivational perspective as a novelty, corroborate the need to interpret the adaptive or maladaptive nature of the task-persistence pattern within contextual models of psychological flexibility [62]. Future lines of research should be directed towards the study of the relevant contextual variables.

In the models performed, preference for fatigue-avoidance goals was always related to activity patterns (activity avoidance and task-contingent persistence), confirming our hypotheses and proving the external validity of this GPQ-F subscale. On the contrary, the Mood-management goal GPQ-F subscale only showed a direct path to pain intensity, not displaying any

significant relationship with the activity patterns. This finding supports the results of the GPQ for pain versions [20, 23]. The set of results obtained suggest, from motivational perspectives, for reasons of parsimony, the only use of the fatigue-avoidance subscale of the GPQ.

The present study has some limitations that should be taken into account. First of all, this is a cross-sectional study and therefore, can only highlight relationships and possible causal paths between the variables analyzed. In the future, new prospective studies should test the results found. Although, the GPQ-F is a contextual measure that overcomes the difficulties that many general instruments have, it is still a self-report measure. Consequently, it shares its limitations, as do the rest of the instruments used in this study. In addition, it must be taken into account that the administration of the GPQ questionnaire, both in the original version and in the Spanish version, due to the type of items raised, may lead to biases in its interpretation. Certain sociodemographic variables (gender, age, educational level, work activity . . .) can affect its completion. This fact is especially relevant considering the wide variability observed in these patients [63]. Along the same lines, obtaining the sample from FM associations and not from primary or specialized healthcare settings may affect the generalization of the results, although it is true that associationism is a very common practice in these patients [64]. Another limitation to bear in mind is that the sample was composed only by women with FM and, as a result, the study findings may not generalize to men or other populations with fatigue.

Along with the limitations, the study shows some noteworthy strengths. First of all, the GPQ-F version created may help to further explore the conflict of goals related to fatigue. This is one of the few studies devoted to understanding fatigue better, a very impairing but forgotten symptom in the context of chronic pain [24]. Likewise, the study has a good sample size, especially compared to other studies with clinical samples.

Future studies should explore the relationships between goal preferences in different clusters of patients with fatigue. Following the results found by Esteve et al. [52], there may be different models explaining the relationship between goal preference and health outcomes in different subgroups of patients. Moreover, pain and also fatigue could have an additional role influencing the relationship between goal preference and activity patterns [62]. Their results show that, on days of greater pain and fatigue, women with FM reported an increase in goal barriers and decreases in goal efforts and progress [62].

## Conclusions

As a conclusion, the study shows initial findings suggesting that the GPQ-F could be a useful tool to analyze goal preferences related to fatigue in clinical samples. The results support the mediational role of activity avoidance patterns in the relationship between preference for fatigue-avoidance goals and fatigue. In this sense, the present study deepens the knowledge regarding the role of fatigue and related goals to shed light onto an understudied area of FM.

## Supporting information

**S1 File. Goal Pursuit Questionnaire Fatiga (GPQ-F).**
(PDF)

**S1 Appendix. Data reporting.**
(SAV)

## Author Contributions

**Conceptualization:** Cecilia Peñacoba, Irene López-Gómez, Maria Angeles Pastor-Mira, Sofía López-Roig, Carmen Ecija.

**Data curation:** Cecilia Peñacoba, Irene López-Gómez, Carmen Ecija.

**Formal analysis:** Cecilia Peñacoba, Irene López-Gómez, Maria Angeles Pastor-Mira, Sofía López-Roig, Carmen Ecija.

**Funding acquisition:** Cecilia Peñacoba.

**Investigation:** Irene López-Gómez, Maria Angeles Pastor-Mira, Sofía López-Roig, Carmen Ecija.

**Methodology:** Cecilia Peñacoba, Irene López-Gómez, Maria Angeles Pastor-Mira, Sofía López-Roig, Carmen Ecija.

**Project administration:** Cecilia Peñacoba.

**Software:** Irene López-Gómez, Carmen Ecija.

**Supervision:** Cecilia Peñacoba.

**Validation:** Maria Angeles Pastor-Mira, Sofía López-Roig.

**Writing – original draft:** Cecilia Peñacoba, Irene López-Gómez, Maria Angeles Pastor-Mira, Sofía López-Roig, Carmen Ecija.

**Writing – review & editing:** Cecilia Peñacoba.

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
