## [Decision Letter · Decision Letter 0]

28 Apr 2021

PONE-D-21-03620

Contextualizing goal preferences in fear-avoidance models. Looking at fatigue as a disabling symptom in fibromyalgia patients.

PLOS ONE

Dear Dr. Peñacoba,

Thank you for submitting your manuscript to PLOS ONE. After careful consideration, we feel that it has merit but does not fully meet PLOS ONE’s publication criteria as it currently stands. Therefore, we invite you to submit a revised version of the manuscript that addresses the points raised during the review process.

I would welcome the submission of a revised manuscript that addresses the major and minor concerns raised in this review. As we explain to all authors, whenever there is a major revision, there is the chance that the new information may reveal fatal flaws that cannot be fixed and that your revised version will not be accepted. We will make every effort to send the revised paper to the same reviewers.

Thank you for considering our Journal. We look forward to seeing this paper again in revised form.

We look forward to receiving your revised manuscript.

Kind regards,

Juan V. Luciano, Ph.D.

Academic Editor

PLOS ONE

Journal Requirements:

2. In your Data Availability statement, you have indicated the following data availability statement ("All relevant data are within the manuscript and its Supporting Information files."), but no such data is available in the submission. PLOS defines a study's minimal data set as the underlying data used to reach the conclusions drawn in the manuscript and any additional data required to replicate the reported study findings in their entirety. All PLOS journals require that the minimal data set be made fully available. For more information about our data policy, please see http://journals.plos.org/plosone/s/data-availability.

5. We noticed you have some minor occurrence of overlapping text with the following previous publication(s), which needs to be addressed:

https://www.frontiersin.org/articles/10.3389/fpsyg.2019.01912/full

The text that needs to be addressed is within your results section.

In your revision ensure you cite all your sources (including your own works), and quote or rephrase any duplicated text outside the methods section. Further consideration is dependent on these concerns being addressed.

Reviewers' comments:

Reviewer's Responses to Questions

**Comments to the Author**

1. Is the manuscript technically sound, and do the data support the conclusions?

Reviewer #1: Yes

Reviewer #2: Yes

2. Has the statistical analysis been performed appropriately and rigorously? 

Reviewer #1: Yes

Reviewer #2: Yes

3. Have the authors made all data underlying the findings in their manuscript fully available?

Reviewer #1: Yes

Reviewer #2: Yes

4. Is the manuscript presented in an intelligible fashion and written in standard English?

Reviewer #1: Yes

Reviewer #2: Yes

5. Review Comments to the Author

Reviewer #1: Authors have written a very clear, valuable and extensive paper. I congratulate them on this piece of work. I have minor comments for the authors to consider.

Pg 8

Ln177: You stated that participants were recruited from different patients’ associations and that mean time since establishing FM diagnosis was 12.14±8.45 years. It is possible, therefore, that certain patients did not fulfill criteria for FM diagnosis at the moment of testing. Please explain how this fact could have influenced your results and consider adding this in limitation section if necessary.

Pg 13

Ln298: Did you perform CFA on the same sample as for the EFA? Please, explain.

Pg15

Ln340: When you report decimal values, sometimes you omit the zero and sometimes you do not. For example “.9” vs “0.9”.

Please revise and be consistent when reporting these values.

Reviewer #2: The objective of the study is interesting from a scientific point of view and is well planned and executed with the chosen methodology. Investigate the knowledge about the relationship between hedonic goals (fatigue-avoidance and mood-management goals) and fatigue, pain and impact of fybromialgia, focusing in fatigue is very interesting, not only from the point of view of reinforcing knowledge about it. condition, but also very interesting from a clinical point of view.

Minor style issues:

To try to reduce space, a common Figure could be made to explain both the “Tested structural model with activity avoidance pattern” and the “Tested structural model with task contingent persistence pattern”. Or results Tables (Figure 4a to 4f)

Figure 2 and 3 appear in text (322) before Figure 1 (378). Figure 1 is not named in text.

Also, Table 2 appears before in text (338) that Table 1 (354)

Despite being an adaptation of the accepted and validated previous questionnaire (GPQ), due to characteristics of the sample, the items are very associated to sociodemographics variables of these sample. Probably no near of reality of all fybromialgya population (mean = 56.9); majority with partner; low studies level; 32 retired (cause of fybromyalgia, probably and also age).

This sample is not so much heterogeneous within the fybromyalgia condition, leading to the construction of a questionnaire that is difficult to understand, apply, and even justify for other profiles of women with fibromyalgia. Activities limited to (clean of house, clean Windows, shopping, sewing ..).

Maybe to see the complete questionnaire could help to review how it finally looks for its application to users.

It is possible that from a statistical point of view the questionnaire is correct, however, from a broader perspective, the questionnaire could be debatable for women as a gender and might not be appropriate from a political correctness point of view.

Maybe the selection of the sample could have been carried out in primary or specialized health face, to try to collect a representative sample (more initial stages of women with fibromyalgia, active at work, with a different sociocultural profile ..)

6. PLOS authors have the option to publish the peer review history of their article (what does this mean?). If published, this will include your full peer review and any attached files.

Reviewer #1: No

Reviewer #2: No

---

## [Author Response · Author response to Decision Letter 0]

13 Jun 2021

Juan V. Luciano, Ph.D.

Academic Editor

PLOS ONE

Dear Dr Luciano, 

We would like to thank you for your interest in our manuscript entitled "Contextualizing goal preferences in fear-avoidance models. Looking at fatigue as a disabling symptom in fibromyalgia patients” (ID: PONE-D-21-03620). We appreciate the time that you and the other reviewers have dedicated to reading the manuscript and providing suggestions. Your suggestions have enriched the manuscript considerably. Likewise, we have incorporated all the comments suggested. Following your directions, we have proceeded to revise our manuscript, highlighting the changes by using the track changes mode in MS Word. 

At the end of this letter, you will find an explanation of the changes made to the manuscript in accordance with your comments. 

Once again, we wish to express our appreciation for the clear improvement of the article made possible by the reviewer and editor’s contributions. We hope the new changes meet their expectations, and we hope that they consider the work apt for publication in PLOS ONE.

Please do not hesitate to suggest any further changes. We are at your disposal for anything else you may require.

Best regards,

Journal Requirements:

Response: Thanks for the comment. In this new revised version that we are now sending we have followed the guide for authors established by Plos one Journal. We have named the files according to the style requirements of the journal.

2. In your Data Availability statement, you have indicated the following data availability statement ("All relevant data are within the manuscript and its Supporting Information files."), but no such data is available in the submission. PLOS defines a study's minimal data set as the underlying data used to reach the conclusions drawn in the manuscript and any additional data required to replicate the reported study findings in their entirety. All PLOS journals require that the minimal data set be made fully available. For more information about our data policy, please see http://journals.plos.org/plosone/s/data-availability.

Response: A underlying data set of our study has been uploaded as “Supporting Information files”: “S2 Appendix. Data reporting”.

Response: The information has been reviewed. The correct information is: “This study was funded by the Health Research Fund (Fondo de Investigación en Salud) from the Instituto de Salud Carlos III (Spain) co-financed by the European Union through the Fondo Europeo de Desarrollo Regional (FEDER), Grant Number PI17/00858”.

Response: Thanks. Now, the ethics statement only appear in the Methods section of our manuscript: “The study was approved by the Bioethics Committee of Rey Juan Carlos University (Reference 160520165916; PI17/00858) and all participants signed an informed consent form to take part in the project”.

5. We noticed you have some minor occurrence of overlapping text with the following previous publication(s), which needs to be addressed:

https://www.frontiersin.org/articles/10.3389/fpsyg.2019.01912/full

The text that needs to be addressed is within your results section.

In your revision ensure you cite all your sources (including your own works), and quote or rephrase any duplicated text outside the methods section. Further consideration is dependent on these concerns being addressed.

Response: Thank you for your comment. The wording of the text has been revised and modified to avoid overlaps.

Reviewer #1 Comments:

Reviewer #1: Authors have written a very clear, valuable and extensive paper. I congratulate them on this piece of work. I have minor comments for the authors to consider.

Response: We appreciate the reviewer’s kind words and their positive appraisal of our study. Following is an explanation of the issues the reviewer would like to have changed or included.

Pg 8

Ln177: You stated that participants were recruited from different patients’ associations and that mean time since establishing FM diagnosis was 12.14±8.45 years. It is possible, therefore, that certain patients did not fulfill criteria for FM diagnosis at the moment of testing. Please explain how this fact could have influenced your results and consider adding this in limitation section if necessary.

Response: We regret the error caused by poor writing of the manuscript. All participants had been diagnosed with fibromyalgia prior to the study. In fact, being diagnosed with fibromyalgia is a mandatory requirement to belong to fibromyalgia associations. In the text we refer to the time elapsed since the patients had been diagnosed. This question has been clarified in the manuscript. The range of years elapsed since the diagnosis and additional information on the recruitment procedure have also been included.

Pg 13

Ln298: Did you perform CFA on the same sample as for the EFA? Please, explain.

Response: We thank the reviewer for highlighting this issue. We did perform the EFA and CFA on the same sample and, after reviewing the literature, we have concluded that it is not convenient. Since the Spanish GPQ-F is a new instrument, even though it is based on the Spanish GPQ (Pastor-Mira et al., 2019), we propose to include only the EFA in the article, along with the structural equation models. The manuscript has been restructured according to the reviewer's suggestion.

Pg15

Ln340: When you report decimal values, sometimes you omit the zero and sometimes you do not. For example “.9” vs “0.9”.

Please revise and be consistent when reporting these values.

Response: We thank the reviewer for his suggestion. The entire manuscript has been reviewed. In all cases, zeros have been used (for example 0.9).

Reviewer #2 Comments:

Reviewer #2: The objective of the study is interesting from a scientific point of view and is well planned and executed with the chosen methodology. Investigate the knowledge about the relationship between hedonic goals (fatigue-avoidance and mood-management goals) and fatigue, pain and impact of fybromialgia, focusing in fatigue is very interesting, not only from the point of view of reinforcing knowledge about it. condition, but also very interesting from a clinical point of view.

Response: We appreciate the reviewer’s kind words and their positive appraisal of our study. Following is an explanation of the issues the reviewer would like to have changed or included.

Minor style issues:

To try to reduce space, a common Figure could be made to explain both the “Tested structural model with activity avoidance pattern” and the “Tested structural model with task contingent persistence pattern”. Or results Tables (Figure 4a to 4f)

Response: Thanks for your wise suggestion. We have proceeded to make a common figure to explain both the “Tested structural model with activity avoidance pattern” and the “Tested structural model with task contingent persistence pattern” (now called “Fig 1. Tested structural model with activity patterns (activity avoidance and task contingent persistence”). The manuscript has been restructured according to the new numbering of the figures

Figure 2 and 3 appear in text (322) before Figure 1 (378). Figure 1 is not named in text. Also, Table 2 appears before in text (338) that Table 1 (354).

Response: Thank you for your comment. The commented errors have been corrected and both the Tables and Figures appear both inserted and mentioned throughout the manuscript in the proper order.

Despite being an adaptation of the accepted and validated previous questionnaire (GPQ), due to characteristics of the sample, the items are very associated to sociodemographics variables of these sample. Probably no near of reality of all fybromialgya population (mean = 56.9); majority with partner; low studies level; 32 retired (cause of fybromyalgia, probably and also age).

This sample is not so much heterogeneous within the fybromyalgia condition, leading to the construction of a questionnaire that is difficult to understand, apply, and even justify for other profiles of women with fibromyalgia. Activities limited to (clean of house, clean Windows, shopping, sewing ..).

Maybe to see the complete questionnaire could help to review how it finally looks for its application to users.

It is possible that from a statistical point of view the questionnaire is correct, however, from a broader perspective, the questionnaire could be debatable for women as a gender and might not be appropriate from a political correctness point of view.

Maybe the selection of the sample could have been carried out in primary or specialized health face, to try to collect a representative sample (more initial stages of women with fibromyalgia, active at work, with a different sociocultural profile ..)

Response: Thanks. We fully share the reviewer's suggestion. Indeed, from a broad perspective, the questionnaire may contain numerous gender, cultural and social biases. As the reviewer points out, in our work we started from a previously proposed and validated instrument in musculoskeletal pain patients in Holland. To minimize these biases and ensure greater adaptation to our population, previously, we adapted the Spanish GPQ for fatigue symptoms with three sub-studies (group structured interview, self-administration questionnaire and thinking-aloud; n = 15-27 patients). In addition, some previous studies by the research team, recently published, have helped us in the "adjustment" of the instrument to the population under study: fibromyalgia.

On the one hand, we performed a validation of the scale in women with fibromyalgia in the Spanish population, taking pain as the main symptom (in the same way that the authors do with the original scale). On the other hand, and given that our objective was to get closer to the symptom of fatigue (probably the most forgotten symptom of fibromyalgia), we carried out a qualitative study to get closer and really understand the meaning and implications of this symptom for the patients. These two previous studies helped us to contextualize the adaptation of this instrument. In any case, given the interesting contribution of the reviewer, we have proceeded to reflect on these biases and their implications on the limitations of the study, in the discussion section. Likewise, following the reviewer's suggestion, we have proceeded to incorporate the Spanish version of the validated questionnaire as a S1 File. Goal Pursuit Questionnaire Fatiga (GPQ-F) to this manuscript. We appreciate the comment again.

Response: Thanks. We used PACE digital diagnostic tool in our figures.

Response: Thanks. We have proceeded to include our updated statement in our cover letter: “This study was funded by the Health Research Fund (Fondo de Investigación en Salud) from the Instituto de Salud Carlos III (Spain) co-financed by the European Union through the Fondo Europeo de Desarrollo Regional (FEDER), Grant Number PI17/00858”.

---

## [Editor Report · Decision Letter 1]

23 Jun 2021

Contextualizing goal preferences in fear-avoidance models. Looking at fatigue as a disabling symptom in fibromyalgia patients.

PONE-D-21-03620R1

Dear Dr. Peñacoba,

We’re pleased to inform you that your manuscript has been judged scientifically suitable for publication and will be formally accepted for publication once it meets all outstanding technical requirements.

Kind regards,

Juan V. Luciano, Ph.D.

Academic Editor

PLOS ONE

Additional Editor Comments (optional):

Dear Dr. Peñacoba:

Thank you for submitting your revised manuscript to Plos One. I have now completed my review, and I am pleased to be able to accept your manuscript in its current form.

I sincerely believe that you have done a thorough job in trying to address the issues raised by both reviewers with regards to your initial submission. After my own reading, no further elaboration and/or commenting seems necessary.

Congratulations for you hard work along the review process.

Sincerely,

Dr. Luciano
---

## [Editor Report · Acceptance letter]

28 Jun 2021

PONE-D-21-03620R1 

Contextualizing goal preferences in fear-avoidance models. Looking at fatigue as a disabling symptom in fibromyalgia patients. 

Dear Dr. Peñacoba:

I'm pleased to inform you that your manuscript has been deemed suitable for publication in PLOS ONE. Congratulations! Your manuscript is now with our production department. 

Kind regards, 

on behalf of

Dr. Juan V. Luciano 

Academic Editor

PLOS ONE